# HAWQ-V2: Hessian Aware trace-Weighted Quantization of Neural Networks

**Zhen Dong, Zhewei Yao, Daiyaan Arfeen,**
**Amir Gholami, Michael W. Mahoney, Kurt Keutzer**
University of California, Berkeley,
{zhendong, zheweiy, daiyaanarfeen, amirgh, mahoneymw, and keutzer}@berkeley.edu

## Abstract

Quantization is an effective method for reducing memory footprint and inference time of Neural Networks. However, ultra low precision quantization could lead to significant degradation in model accuracy. A promising method to address this is to perform mixed-precision quantization, where more sensitive layers are kept at higher precision. However, the search space for a mixed-precision quantization is exponential in the number of layers. Recent work has proposed a novel Hessian based framework [9], with the aim of reducing this exponential search space by using second-order information. While promising, this prior work has three major limitations: (i) they only use a heuristic metric based on top Hessian eigenvalue as a measure of sensitivity and do not consider the rest of the Hessian spectrum; (ii) their approach only provides relative sensitivity of different layers and therefore requires a manual selection of the mixed-precision setting; and (iii) they do not consider mixed-precision activation quantization. Here, we present HAWQ-V2 which addresses these shortcomings. For (i), we theoretically prove that the right sensitivity metric is the average Hessian trace, instead of just top Hessian eigenvalue. For (ii), we develop a Pareto frontier based method for automatic bit precision selection of different layers without any manual intervention. For (iii), we develop the first Hessian based analysis for mixed-precision activation quantization, which is very beneficial for object detection. We show that HAWQ-V2 achieves new state-of-the-art results for a wide range of tasks. In particular, we present quantization results for InceptionV3 (7.57MB with $75.98\%$ accuracy), ResNet50 (7.99MB with $75.92\%$ accuracy), and SqueezeNext (1MB with $68.68\%$ accuracy), all without any manual bit selection. Furthermore, we present results for object detection on Microsoft COCO, where we achieve 2.6 higher mAP than direct uniform quantization and 1.6 higher mAP than the recently proposed method of FQN, with a smaller model size of 17.9MB.

## 1 Introduction

Deep convolutional Neural Networks (NNs) have achieved great success in recent years. However, many of these models, particularly those with state-of-the-art performance, have a high computational cost and memory footprint. This slows inference and training in the cloud, and prohibits their deployment on edge devices.

Quantization [1, 6, 15, 36, 34, 17, 35, 8, 32, 33, 29, 9, 10, 5, 7] is a very promising approach to address this problem by reducing the memory bottleneck, thus allowing the use of lower precision computational units in hardware. By replacing floating point weights in the model with low precision fixed-point values, quantization can shrink the model size without changing the original network architecture. The gains in speed and power consumption directly depend on how aggressively we

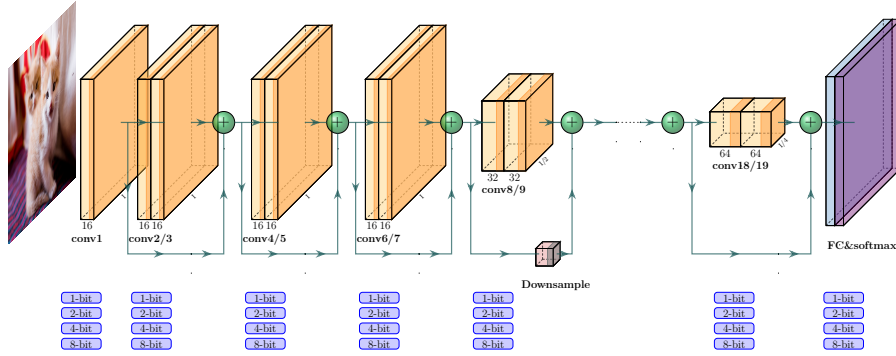

**Figure 1:** *Mixed Precision Illustration of ResNet20. Here we show the network architecture and list four possible bit precision setting for each layer.*

can perform quantization without losing generalization/accuracy of the model. Despite significant advances, performing ultra low-bit quantization results in non-trivial degradation on accuracy.

Notable recent work on quantization includes using non-uniform quantizers [32, 19], progressive quantization-aware fine-tuning [34, 9], and mixed-precision quantization [30, 29, 9, 27]. Despite the use of non-uniform quantization (which is generally difficult for efficient implementation on hardware), the accuracy degradation is still significant for ultra-low precision. A promising approach to address this is through mixed-precision quantization, where some layers are kept at higher precision, and other layers at lower precision. However, a major problem with this approach is that the search space for determining a good mixed-precision quantization setting is exponentially large in the number of NN layers. This is schematically shown in Figure 1, where four possible bit settings of 1/2/4/8 are considered for each layer of ResNet20. Finding a mixed-precision setting using these possible bitwidths, has a search space of size $4^{20} \approx 1 \times 10^{12}$. It is computationally impossible to test all of these mixed-precision settings and choose a particular setting with good generalization and good hardware performance (in terms of latency and power). The recent work of [29] proposed a reinforcement learning based method to address this. However, this consumes orders of magnitude larger time for bit precision selection (Table 4). Another notable approach is differentiable neural architecture search (DNAS) based methods [30]. But these searching methods can require a large amount of computational resources, and the quality of quantization is very sensitive to the initialization of their search parameters and therefore unpredictable. This makes the deployment of these methods in online learning scenarios especially challenging, as in these applications a new model is trained every few hours and needs to be quantized for efficient inference.

To address these issues, recent work introduced HAWQ [9], which aims to assign higher bit-precision to layers that are more sensitive, and lower bit-precision to less sensitive layers. The sensitivity is measured based on the top Hessian eigenvalue of each layer. This can reduce the exponential search space for mixed-precision quantization, since a layer with higher Hessian eigenvalues cannot be assigned lower bits, as compared to another layer with smaller Hessian eigenvalues. However, there are several shortcomings of this approach: (i) HAWQ uses a heuristic metric based on top Hessian eigenvalue as a measure of sensitivity, and it ignores the rest of the Hessian spectrum; (ii) HAWQ only provides relative sensitivity of different layers, and it still requires a **manual** selection of the mixed-precision setting; and (iii) HAWQ does not consider mixed-precision activation quantization.

Here, we address these challenges and introduce the HAWQ-V2 method. Our main contributions are the following.

1. We perform a theoretical analysis (Lemma 1) showing that a better sensitivity metric is the average Hessian trace, instead of the top eigenvalue heuristic used in HAWQ [9].
2. The HAWQ framework [9] only provides relative sensitivity, and thus it requires manual intervention to select the precise bit-precision setting for each layer. We address this by using a Pareto-frontier based method to automatically determine the bit precision of different layers without any manual selection (Figure 4).
3. We implement a fast algorithm to compute the Hessian trace information using Hutchinson's algorithm in PyTorch. A common concern with the application of Hessian-based methods is the

computational cost [31], but we show this is not an issue here. For example, we can compute Hessian trace for all 54 layers in ResNet50 in less than 30 minutes with 4 GPUs (only 33s per block on average). This is quite fast compared to manual mixed-precision methods which often take weeks of tuning.

4. We achieve new state-of-the-art results for a wide range of models. We present quantization results for InceptionV3, ResNet50, and SqueezeNext (Table 1). Furthermore, we present results for object detection on the Microsoft COCO dataset, where HAWQ-V2 achieves 2.6 higher mAP than direct uniform quantization and 1.6 higher mAP than the recently proposed method of FQN [18], with even smaller model size 17.9MB (Table 2).

5. We develop mixed-precision activation quantization, as described in §2.2. We propose a fast method for computing Hessian information w.r.t. activations, and we show that mixed-precision activation can boost the performance of our object detection model to 34.4 mAP (Table 2).

*Outline:* In § 2, we discuss theoretical analysis and the relationship between the Hessian spectrum and quantization; and we then discuss the Pareto frontier and our automatic precision selection method. Then, in § 3, we show the results of the trade-off between speed and convergence in the Hutchinson algorithm; and we test HAWQ-V2 with various models on both image classification and object detection tasks. Finally, we provide a conclusion in § 4.

## 2    Methodology

For a supervised learning framework, the goal is to minimize the empirical risk loss,

$$\mathcal{L}(\theta) = \frac{1}{N} \sum_{i=1}^{N} f(x_i, y_i, \theta),$$  (1)

where $\theta \in R^d$ is the learnable model parameters, and $f(x, y, \theta)$ is the loss for a datum $(x, y) \in (X, Y)$. Here, $N = |X|$ is the cardinality of the training set. Assume that the NN can be partitioned into $L$ layers as $\{B_1, B_2, \cdots, B_L\}$, with corresponding learnable parameters $\{W_1, W_2, \cdots, W_L\}$. Furthermore, we denote mini-batch gradient of the loss w.r.t. model parameters as $g = \frac{1}{N_B} \sum_{i=1}^{N_B} \frac{\partial f}{\partial \theta}$, and sub-sampled Hessian w.r.t. parameters as $H = \frac{1}{N_B} \sum_{i=1}^{N_B} \frac{\partial^2 f}{\partial \theta^2}$, where $N_B$ is the mini-batch.

### 2.1    Sensitivity Metric

HAWQ uses the top Hessian eigenvalue to determine the relative sensitivity order of different layers [9]. However, a NN model contains millions of parameters, and thus millions of Hessian eigenvalues. Therefore, just measuring the top eigenvalue can be sub-optimal. As a simple example, consider two functions $F_1(x, y) = 100x^2 + y^2$ and $F_2(x, y) = 100x^2 + 99y^2$. The top Hessian eigenvalues of $F_1$ and $F_2$ are the same (i.e., 200). However, it is clear that $F_2$ is more sensitive than $F_1$ since $F_2$ has much larger function value change along y-axis. Below, we perform a theoretical analysis and show that a better metric is to compute the average Hessian trace (i.e., average of all Hessian eigenvalues) instead of just the top eigenvalue, and later in Section 3.4 we perform an empirical ablation study which supports this finding. Note that in practice the trace and top eigenvalues can be significantly different, as shown in Figure 6 in Appendix C.

**Assumption 1** *Assume that:*

- *The model is twice differentiable and has converged to a local minima such that the first and second order optimality conditions are satisfied, i.e., the gradient is zero and the Hessian is positive semi-definite.*
- *If we denote the Hessian of the $i^{th}$ layer as $H_i$, and its corresponding orthonormal eigenvectors as $v_1^i, v_2^i, ..., v_{n_i}^i$, then the quantization-aware fine-tuning perturbation, $\Delta W_i^* = \arg\min_{W_i^* + \Delta W_i^* \in Q(\cdot)} L(W_i^* + \Delta W_i^*)$, satisfies*

$$\Delta W_i^* = \alpha_{bit} v_1^i + \alpha_{bit} v_2^i + ... + \alpha_{bit} v_{n_i}^i.$$  (2)

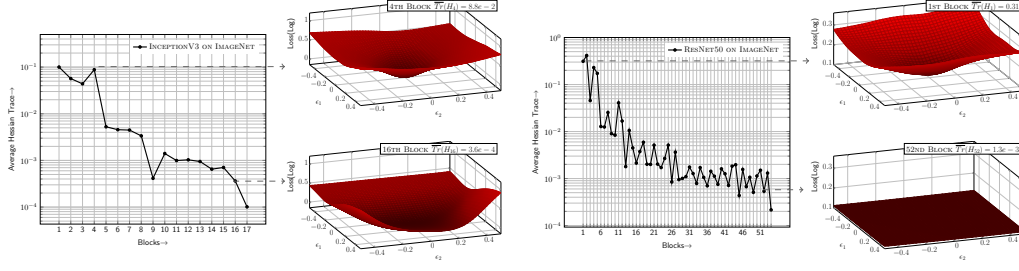

**Figure 2:** *Average Hessian trace of different blocks in InceptionV3 and ResNet50 on ImageNet, along with the loss landscape of the block 4 and 16 in InceptionV3 (block 1 and 52 in ResNet50). As one can see, the average Hessian trace is significantly different for different blocks.*

> Here, $n_i$ is the dimension of $W_i$, $W_i^*$ is the converging point of $i^{th}$ layer, and $Q(\cdot)$ is the quantization function which maps floating point values to reduced precision values. Note that $\alpha_{bit}$ is a constant number based on the precision setting and quantization range.[1]

- *The third-order term, $\|\frac{\nabla^3 \mathcal{L}}{\nabla W_i^3}\| \, \|\Delta W_i^*\|^3/6$ in the Taylor expansion series is small.*

Given this assumption, we establish the following lemma.

**Lemma 1** *Under Assumption 1, when we quantize two layers (denoted by $B_1$ and $B_2$) with same amount of perturbation, namely $\|\Delta W_1^*\|_2^2 = \|\Delta W_2^*\|_2^2$, we will have:*

$$\mathcal{L}(W_1^* + \Delta W_1^*, W_2^*, \cdots, W_L^*) \leq \mathcal{L}(W_1^*, W_2^* + \Delta W_2^*, W_3^*, \cdots, W_L^*), \qquad (3)$$

*if*

$$\frac{1}{n_1} Tr(\nabla_{W_1}^2 \mathcal{L}(W_1^*)) \leq \frac{1}{n_2} Tr(\nabla_{W_2}^2 \mathcal{L}(W_2^*)). \qquad (4)$$

**Proof** Denote the gradient and Hessian of the first layer as $g_1$ and $H_1$, correspondingly. By Taylor's expansion we have:

$$\mathcal{L}(W_1^* + \Delta W_1^*) = \mathcal{L}(W_1^*) + g_1^T \Delta W_1^* + \frac{1}{2} \Delta W_1^{*T} H_1 \Delta W_1^* = \mathcal{L}(W_1^*) + \frac{1}{2} \Delta W_1^{*T} H_1 \Delta W_1^*.$$

Here, we have used the fact that the gradient at the optimum point is zero and that the loss function is locally convex. Also note that $\mathcal{L}(W_1^*) = \mathcal{L}(W_2^*)$ since the model has the same loss before we quantize any layer. Based on the assumption, $\Delta W_1^*$ can be decomposed by the eigenvectors of the Hessian. As a result we have:

$$\Delta W_1^{*T} H_1 \Delta W_1^* = \sum_{i=1}^{n_1} \alpha_{bit,1}^2 v_i^{1T} H_1 v_i^1 = \alpha_{bit,1}^2 \sum_{i=1}^{n_1} \lambda_i^1,$$

where $(\lambda_i^1, v_i^1)$ is the corresponding eigenvalue and eigenvector of Hessian. Similarly, for the second layer we will have: $\Delta W_2^{*T} H_2 \Delta W_2^* = \alpha_{bit,2}^2 \sum_{i=1}^{n_2} \lambda_i^2$, where $\lambda_i^2$ is the $i^{th}$ eigenvalue of $H_2$. Since $\|\Delta W_1^*\|_2 = \|\Delta W_2^*\|_2$, we have $\sqrt{n_1} \alpha_{bit,1} = \sqrt{n_2} \alpha_{bit,2}$. Therefore, we have:

$$\mathcal{L}(W_2^* + \Delta W_2^*) - \mathcal{L}(W_1^* + \Delta W_1^*) = \alpha_{bit,2}^2 n_2 (\frac{1}{n_2} \sum_{i=1}^{n_2} \lambda_i^2 - \frac{1}{n_1} \sum_{i=1}^{n_1} \lambda_i^1) \geq 0.$$

It is easy to see that the lemma holds since the sum of eigenvalues equals to the trace of the matrix. □

It should be noted that the proof still holds for cases where $\|\Delta W_1^*\|_2^2 \neq \|\Delta W_2^*\|_2^2$. In such cases, Eq. 4 becomes:

$$\frac{\|\Delta W_1^*\|_2^2}{n_1} Tr(\nabla_{W_1}^2 \mathcal{L}(W_1^*)) \leq \frac{\|\Delta W_2^*\|_2^2}{n_2} Tr(\nabla_{W_2}^2 \mathcal{L}(W_2^*)), \qquad (5)$$

indicating that $\overline{Tr}(H_i)\|\Delta W_i^*\|_2^2$ can be used as a measure of sensitivity.

At first, computing the Hessian trace may seem a prohibitive task, as we do not have direct access to the elements of the Hessian matrix. Furthermore, forming the Hessian matrix explicitly is not

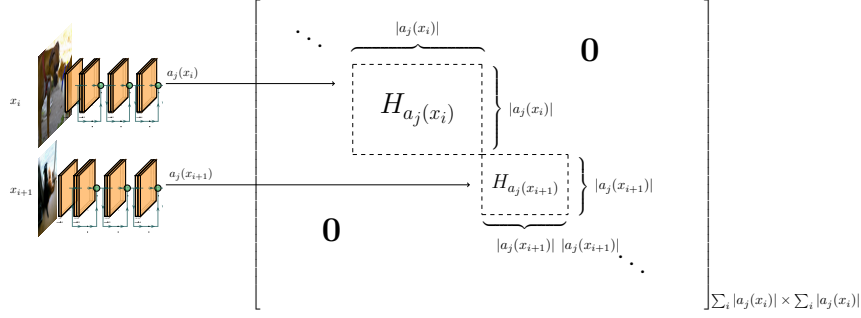

**Figure 3:** *Illustration of the structure of Hessian w.r.t to activations ($H_{a_j}$). It is evident that different sized inputs $x_i$ will produce different sized blocks $H_{a_j(x_i)}$ which appear on the diagonal of $H_{a_j}$.*

computationally feasible. However, it is possible to leverage the extensive literature in Randomized Numerical Linear Algebra (RandNLA) [22, 23] which address this type of problem. In particular, the seminar works of [2, 3] have proposed randomized algorithms for fast trace estimation, using so-called matrix-free methods which do not require the explicit formation of the Hessian operator. Here, we are interested in the trace of a symmetric matrix $H \in R^{d \times d}$. Then, given a random vector $z \in R^d$ whose component is i.i.d. sampled Gaussian distribution ($N(0,1)$) (or Rademacher distribution), we have:

$$Tr(H) = Tr(HI) = Tr(H\,\mathbb{E}[zz^T]) = \mathbb{E}[Tr(Hzz^T)] = \mathbb{E}[z^T Hz], \qquad (6)$$

where $I$ is the identity matrix. Based on this, the Hutchinson algorithm [2] can be used to estimate the Hessian trace:

$$Tr(H) \approx \frac{1}{m}\sum_{i=1}^{m} z_i^T H z_i = Tr_{Est}(H). \qquad (7)$$

We show empirically in §3.1 that this algorithm has good convergence properties, resulting in trace computation being orders of magnitude faster than training the network itself.

We have incorporated the above approach and computed the average Hessian trace for different layers of InceptionV3 and ResNet50, as shown in Figure 2. As one can see, there is a significant difference between average Hessian trace for different layers. To better illustrate this, we have also plotted the loss landscape of InceptionV3 and ResNet50 by perturbing the pre-trained model along the first and second eigenvectors of the Hessian for each layer. It is clear that different layers have significantly different "sharpness." (In Appendix C, we also show the average Hessian trace for different blocks of SqueezeNext and RetinaNet, as well as their corresponding loss landscape; see Figure 7.)

## 2.2 Mixed Precision Activation

The above analysis is not restricted to weights, and in fact it can be extended to mixed-precision activation quantization. In § 3, we will show that this is particularly useful for tasks such as object detection. The theoretical results remain the same, except that the Hessian here is with respect to activations instead of model parameters. In the matrix-free Hutchinson algorithm, we need the result of the following Hessian-vector product to compute the Hessian trace:

$$z^T H_{a_j} z = z^T \left( \nabla_{a_j}^2 \frac{1}{N}\sum_{i=1}^{N} f(x_i, y_i, \theta) \right) z, \qquad (8)$$

where $a_j$ is the activations of the $j^{th}$ layer. Here, $H_{a_j} \in \mathbb{R}^{(\sum_{i=1}^N |a_j(x_i)|) \times (\sum_{i=1}^N |a_j(x_i)|)}$, where $|a_j(x_i)|$ is the size of the activation of the $j^{th}$ layer for $i^{th}$ input. This is because $a_j$ is a concatenation of $a_j(x_i), \forall i$. See Figure 3 for illustration of the matrix $H_{a_j}$ and its shape. Not only is it prohibitive to compute Hessian matrix, the Hessian-vector product is also infeasible since even generating the random vectors $z \in \mathbb{R}^{\sum_{i=1}^N |a_j(x_i)|}$ is prohibitive, let alone computing its product with $H_{a_j}$. Furthermore, note that $a_j$ depends on $x_i$, and for many tasks such as object detection on Microsoft COCO, $x_i$ does not have a fixed size. As a result, the activation size of each layer depends on the input data and is not fixed, which further complicates computing Hessian trace w.r.t. activations.

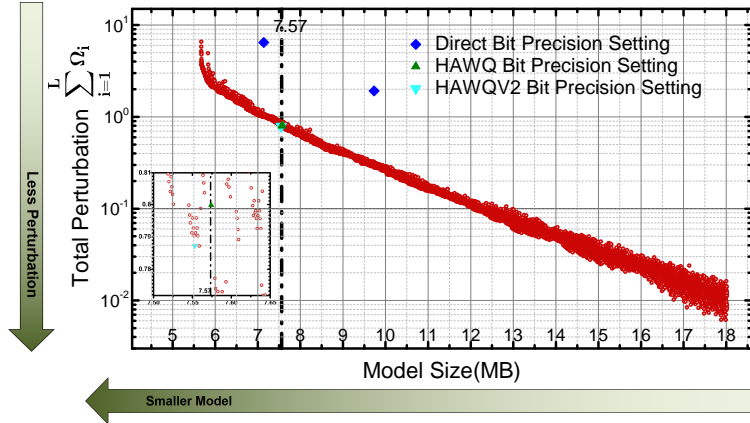

**Figure 4:** *Pareto Frontier: The trade-off between model size and the sum of $\Omega$ metric (of Eqn. (10)) in InceptionV3. Here, L is the number of blocks in the model, and each point in the figure stands for a specific bit precision setting. We show the precision setting used in Direct quantization and* HAWQ. *To achieve fair comparison, we set constraint on* HAWQ-V2 *to have the same model size as* HAWQ.

However, $H_{a_j}$, has a very interesting structure. As illustrated in Figure 3, it is block diagonal, with $H_{a_j(x_i)}$ being the blocks, where $H_{a_j(x_i)} = \nabla^2_{a_j(x_i)} \frac{1}{N} f(x_i, y_i, \theta)$. This is due to the fact that different inputs are independent of each other. As a result, we can compute the Hessian-trace for the layer's activations *for one input* at a time, and then average the resulting Hessian-traces of each block diagonal part, i.e.,

$$z^T H_{a_j} z = \frac{1}{N} \sum_{i=1}^{N} z_i^T H_{a_j(x_i)} z_i, \tag{9}$$

where $z_i$ is the corresponding components of $z$ w.r.t. the $i^{th}$ input, i.e., $x_i$. We note that usually this trace computation converges very fast, and it is not necessary to average over the entire dataset. See Figure 8 in Appendix for more details.

## 2.3 Automatic Bit Selection

An important limitation of relative sensitivity analysis is that it does not provide the specific bit precision setting for different layers. This is true even if we use the average Hessian trace, instead of the top Hessian eigenvalue. For example, we show the average Hessian trace for different blocks of InceptionV3 in Figure 2. We can clearly see that block 1 to block 4 have the largest average Hessian trace, and block 9 or block 16 have orders of magnitude smaller average Hessian trace. However, although we know the first four blocks are more sensitive, we still cannot determine whether to assign 8-bit or 4-bit for these layers.

Denote by $\mathcal{B}$ the set of all admissible bit precision settings that satisfy the relative sensitivity analysis based on the average Hessian trace discussed above. Compared to the original exponential search space, applying the sensitivity constraint makes the cardinality (size) of $\mathcal{B}$ significantly smaller. As an example, the original mixed-precision search space for ResNet50 is $4^{50} \approx 1.3 \times 10^{30}$ if bit-precisions are chosen among $\{1, 2, 4, 8\}$. Using the Hessian-trace sensitivity constraint significantly reduces this search space to $|\mathcal{B}| = 2.3 \times 10^4$ (details on how to calculate the size of $|\mathcal{B}|$ are included in Appendix B). However, this search space is still prohibitively large, especially for deeper models such as ResNet152. In the HAWQ paper [9], the authors manually chose the bit precision among this reduced search space, but this manual selection is undesirable.

We found that this problem can be efficiently addressed using a Pareto frontier approach. The main idea is to sort each candidate bit-precision setting in $\mathcal{B}$ based on the total second-order perturbation that they cause, according to the following metric:

$$\Omega = \sum_{i=1}^{L} \Omega_i = \sum_{i=1}^{L} \overline{Tr}(H_i) \cdot \|Q(W_i) - W_i\|_2^2, \tag{10}$$

where $i$ refers to the $i^{th}$ layer, L is the number of layers in the model, $\overline{Tr}(H_i)$ is the average Hessian trace, and $\|Q(W_i) - W_i\|_2$ is the $L_2$ norm of quantization perturbation. The intuition is that a bit precision setting with minimal second-order perturbation to the model should lead to good generalization after quantization-aware fine-tuning. Given a target model size, we sort the elements of $\mathcal{B}$ based on their $\Omega$ value, and we choose the bit precision setting with minimal $\Omega$. While this approach cannot theoretically guarantee the best possible performance, we have found that in practice it can generate bit precision settings that exceed current state-of-the-art results with a small time cost (as shown in Section 3.1). An important benefit of this approach is that it removes the manual precision selection process used in our previous work on HAWQ [9].

We show the process for choosing the exact bit precision setting of InceptionV3 in Figure 4 (details in Appendix E). Each red dot denotes a specific bit precision setting for different blocks of InceptionV3. For each target model size, HAWQ-V2 chooses the bit precision setting with minimal $\Omega$ value. With green triangles, we have also denoted the bit precision setting that was manually selected in the HAWQ paper [9]. The automatic bit precision setting of HAWQ-V2 exceeds the accuracy of HAWQ, as will be discussed in the next section.

# 3 Empirical Results

## 3.1 Hutchinson's Method for Trace Estimation

In Figure 5, we show the convergence plot for the Hutchinson's algorithm as we increase the number of iterations used for the Hessian trace estimation. It can be clearly seen that the trace converges rapidly as we increase the number of data points over 512, over which the sub-sampled Hessian is computed. We can see that 50 Hutchinson iterations are sufficient to achieve an accurate approximation with low variance. Based on the convergence analysis, we are able to calculate all the average Hessian traces, shown in Figure 2, corresponding to 54 blocks in a ResNet50 model, within 30 minutes (33s per block on average) using 4 GPUs. The Hutchinson algorithm, in addition to the automatic bit precision selection, makes HAWQ-V2 a significantly faster algorithm than previous searching based algorithms [29] (up to $120\times$ faster as shown in Table 4 in Appendix C).

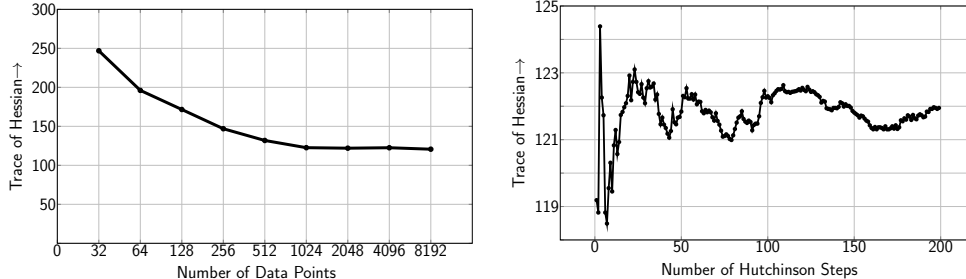

**Figure 5:** *Relationship between the convergence of Hutchinson and the number of data points (Left) as well as the number of steps (Right) used for trace estimation on block 21 in ResNet50.*

## 3.2 ImageNet

As shown in Table 1, we first apply HAWQ-V2 on ResNet50 [14], and compare HAWQ-V2 with other popular quantization methods [35, 8, 32, 13, 29, 9]. It should be noted that [35, 8, 32, 13] followed traditional quantization rules which set the precision of the first and last layers to 8-bit, and quantized other layers to an identical precision. Both [29, 9] are mixed-precision quantization methods. Also, [29] uses reinforcement learning methods to search for a good precision setting, while HAWQ uses second-order information to guide the precision selection as well as the block-wise fine-tuning. HAWQ achieves $75.48\%$ with a 7.96MB model size. Keeping model size the same, HAWQ-V2 can achieve $75.92\%$ accuracy without any heuristic knowledge or manual efforts.

We then show results on InceptionV3 [28]. Direct quantization of InceptionV3 (i.e., without use of second-order information), results in $7.69\%$ accuracy degradation. Using the approach proposed in [16] results in more than $2\%$ accuracy drop, even though it uses higher bit precision. HAWQ [9] results in a $2\%$ accuracy gap with a compression ratio of $12.04\times$. HAWQ-V2 **automatically** generates the exact precision setting for the whole network, and still achieves better accuracy than the manual method of HAWQ.

**Table 1:** *Quantization results on ImageNet. We abbreviate quantization bits used for weights as "w-bits," quantization bits used for activations as "a-bits," top-1 testing accuracy as "Top-1," and weight compression ratio as "W-Comp." Furthermore, we compare HAWQ-V2 with direct quantization method of [9] ("Direct") and other state-of-the-art quantization methods. Here "MP" refers to mixed-precision quantization, and we show the lowest bit-precision used in a mixed-precision setting. Compared to [16, 25], we achieve higher compression ratio with higher testing accuracy.*

(a) ResNet50 on ImageNet.

| Method | w-bits | a-bits | Top-1 | W-Comp | Size(MB) |
|---|---|---|---|---|---|
| Baseline | 32 | 32 | 77.39 | 1.00× | 97.8 |
| Dorefa [35] | 2 | 2 | 67.10 | 16.00× | 6.11 |
| Dorefa [35] | 3 | 3 | 69.90 | 10.67× | 9.17 |
| PACT [8] | 2 | 2 | 72.20 | 16.00× | 6.11 |
| PACT [8] | 3 | 3 | 75.30 | 10.67× | 9.17 |
| LQ-Nets [32] | 3 | 3 | 74.20 | 10.67× | 9.17 |
| Deep Comp. [13] | 3 | MP | 75.10 | 10.41× | 9.36 |
| HAQ [29] | MP | MP | 75.30 | 10.57× | 9.22 |
| HAWQ [9] | 2 MP | 4 MP | 75.48 | 12.28× | 7.96 |
| HAWQ-V2 | 2 MP | 4 MP | **75.92** | 12.24× | 7.99 |

(b) InceptionV3 on ImageNet

| Method | w-bits | a-bits | Top-1 | W-Comp | Size(MB) |
|---|---|---|---|---|---|
| Baseline | 32 | 32 | 77.45 | 1.00× | 91.2 |
| IntOnly [16] | 8 | 8 | 75.40 | 4.00× | 22.8 |
| RVQ [25] | 3 MP | 3 MP | 74.14 | 10.67× | 8.55 |
| Direct [9] | 2 MP | 4 MP | 69.76 | 15.88× | 5.74 |
| HAWQ [9] | 2 MP | 4 MP | 75.52 | 12.04× | 7.57 |
| HAWQ-V2 | 2 MP | 4 MP | **75.98** | 12.04× | 7.57 |

(c) SqueezeNext on ImageNet

| Method | w-bits | a-bits | Top-1 | W-Comp | Size(MB) |
|---|---|---|---|---|---|
| Baseline | 32 | 32 | 69.38 | 1.00× | 10.1 |
| Direct [9] | 3 MP | 8 | 65.39 | 9.04× | 1.12 |
| HAWQ [9] | 3 MP | 8 | 68.02 | 9.26× | 1.09 |
| HAWQ-V2 | 3 MP | 8 | **68.68** | 9.40× | 1.07 |

We also apply HAWQ-V2 to quantize deep and highly compact models such as SqueezeNext [11]. We choose the wider SqueezeNext model which has a baseline accuracy of 69.38% with 2.5 million parameters (10.1MB in single precision). We can see from Table 1 that direct quantization of SqueezeNext (i.e., without use of second-order information), results in 3.98% accuracy degradation. HAWQ results in a 1MB model size, with 1.36% top-1 accuracy drop. By applying HAWQ-V2 on SqueezeNext, we can achieve a 68.68% accuracy with an unprecedented model size of 1.07MB. Furthermore, in Table 4 we show the timing of HAWQ-V2 as compared to [29]. As one can see, despite using second order information, HAWQ-V2 is orders of magnitude faster and results in significantly more accurate models.

## 3.3 Microsoft COCO

In order to show the generalization capability of HAWQ-V2, we also test object detection task Microsoft COCO 2017 [21]. RetinaNet [20] is a single stage detector that can achieve state-of-the-art mAP with a very simple network architecture. As shown in Table 2, we use the pretrained RetinaNet with ResNet50 backbone as our baseline model, which can achieve 35.6 mAP with 145MB model size. We first show the result of direct quantization where no Hessian information is used. Even with quantization-aware fine-tuning and channel-wise quantization of weights, directly quantizing weights and activations in RetinaNet to 4-bit causes a significant 4.1 mAP degradation. FQN [18] is a recently proposed quantization method which reduces this accuracy gap to 3.1 mAP with the same compression ratio as Direct method. We implement HAWQ to perform mixed-precision quantization, which results in 33.5 mAP. However, using HAWQ-V2 achieves a state-of-the-art performance of 34.1 mAP, which is 0.6 mAP higher than [9] and 1.6 mAP higher than [18] with an even smaller model size.

It should also be noted that we found the activation quantization to be sensitive for object detection models. For instance, increasing activation quantization bit precision to 6-bit, can results in a 34.8 mAP. One might argue that using 6-bit for activation leads to higher activation memory, which can be a problem for extreme cases such as on micro-controllers where every bit counts. For these situations, we can use mixed-precision activation as discussed in §2.2, with the same automatic bit-precision selection method using Pareto optimal curve. As can be seen in Table 2, mixed-precision activation quantization can achieve very good trade-off between accuracy and compression. With only marginal change to activation compression ratio, it can achieve 34.4 mAP, which significantly outperforms uniform 4-bit activation, and is even close to a uniform 6-bit activation quantization.

**Table 2:** *Quantization results of RetinaNet-ResNet50 on Microsoft COCO 2017. We show results of direct quantization, mixed-precision quantization [9], as well as a state-of-the-art quantization method for object detection [18].* HAWQ-V2 *can outperform previous results by a large margin. We also show that* HAWQ-V2 *with mixed-precision activations can achieve even better mAP, with a slightly lower activation compression ratio.*

| Method | w-bits | a-bits | mAP | W-Comp | A-Comp | Size(MB) |
|---|---|---|---|---|---|---|
| Baseline | 32 | 32 | 35.6 | $1.00\times$ | $1.00\times$ | 145 |
| Direct | 4 | 4 | 31.5 | $8.00\times$ | $8.00\times$ | 18.13 |
| FQN [18] | 4 | 4 | 32.5 | $8.00\times$ | $8.00\times$ | 18.13 |
| HAWQ | 3 MP | 4 | 33.5 | $8.10\times$ | $8.00\times$ | 17.90 |
| HAWQ-V2 | 3 MP | 4 | **34.1** | $8.10\times$ | $8.00\times$ | 17.90 |
| HAWQ-V2 | 3 MP | 4 MP | **34.4** | $8.10\times$ | $7.62\times$ | 17.90 |
| HAWQ-V2 | 3 MP | 6 | **34.8** | $8.10\times$ | $5.33\times$ | 17.90 |

### 3.4 Ablation Study

Here, we perform three ablation studies. First, we show why it is important to choose the bit-precision setting that results in the smallest model perturbation as done in Figure 4. The results are shown in Table 3(a), where the ablation row uses a bit precision setting with large model perturbation. As one can see, the HAWQ-V2 approach achieves more than 1% higher accuracy with a smaller model size.

Second, we measure the importance of using the Hessian trace to weight the sensitivity $\Omega_i = \overline{Tr}(H_i)\|\Delta W_i\|_2^2$ in Eq. 10. The results are shown in Table 3(b), where we compare with using only parameter perturbation as the sensitivity metric $\Omega_i = \|\Delta W_i\|_2^2$. As we can see, HAWQ-V2 with average Hessian trace is 0.85% better than L2-Sensitivity, while achieving a smaller model size.

Finally, we also compare HAWQ-V2 with a sensitivity that is weighted by Top-1 Hessian eigenvalue. The results are shown in Table 5 in Appendix D. As expected, the trace-weighted metric in HAWQ-V2 achieves higher accuracy.

**Table 3:** *The effectiveness of metric in Eq. 10. Experiments are for SqueezeNext on ImageNet.*

(a) Accuracy v.s. Total Perturbation

| Method | w-bits | a-bits | Top-1 | Size(MB) | Perturb. |
|---|---|---|---|---|---|
| Baseline | 32 | 32 | 69.38 | 10.1 | 0 |
| Large Perturbation (Ablation) | 3 MP | 8 | 67.46 | 1.09 | 3.2 |
| Min Perturbation (HAWQ-V2) | 3 MP | 8 | **68.68** | **1.07** | 1.1 |

(b) $\overline{Tr}(H_i)\|\Delta W_i\|_2^2$ v.s. $\|\Delta W_i\|_2^2$

| Method | w-bits | a-bits | Top-1 | W-Comp | Size(MB) |
|---|---|---|---|---|---|
| Baseline | 32 | 32 | 69.38 | $1.00\times$ | 10.1 |
| L2-Sensitivity (Ablation) | 3 MP | 8 | 67.83 | $9.18\times$ | 1.10 |
| Trace-Sensitivity (HAWQ-V2) | 3 MP | 8 | **68.68** | $9.40\times$ | **1.07** |

## 4 Conclusions

In this work, we performed a theoretical analysis showing that a better sensitivity metric is the average Hessian trace, instead of the top Hessian eigenvalue heuristic used in HAWQ [9], and we presented an automatic mixed-precision quantization method to avoid the manual bit selection. Moreover, we developed mixed-precision activation, and we proposed a very efficient method for computing the Hessian trace by using matrix-free algorithms. HAWQ-V2 achieves state-of-the-art results on image classification for InceptionV3, ResNet50 and SqueezeNext, and on object detection for RetinaNet-ResNet50. As part of future work, one could consider co-designing quantization for a particular hardware and include not only the Hessian but also hardware-specific metrics such as latency/power consumption [29]. Another direction is to avoid using any floating point number, which is typically used to perform arithmetic on hardware with existing methods [16]. Furthermore, one could explore to make training more efficient by incorporating Hessian to adjust precision throughout training [24], or use it to compress communication in distributed training [26, 12].

## Acknowledgments

This work was supported by a gracious fund from Intel corporation, and in particular Intel VLAB team. We also acknowledge gracious support from Google Cloud, Google TFTC team, Samsung SAIT, and Amazon AWS. We also gratefully acknowledge the support of NVIDIA Corporation for their donation of two Titan Xp GPU used for this research. Michael W. Mahoney would also like to acknowledge the UC Berkeley CLTC, ARO, NSF, and ONR for providing partial support of this work. Our conclusions do not necessarily reflect the position or the policy of our sponsors, and no official endorsement should be inferred.

## Broader Impact

Deep learning models are rapidly increasing in size and this has created a challenge for deploying these models in practice. Our work addresses this problem by developing a novel compression method with minimal impact on accuracy. Our work is applicable to a wide range of NN models, as depicted in our empirical evaluation. Furthermore, our method can help reduce both the computation and the memory bottleneck, so that state-of-the-art neural network models can be deployed even onto edge devices. This helps to realize wider applications of technology, especially in area such as deep learning based daily applications on smart phones, or basic and always-on applications on embedded chips.

## Footnotes

[1]We assume $\alpha_{bit}$ a constant for simplicity. It can be relaxed to random coefficients with the same second moment, i.e., $\alpha_{bit}$ can be random variables for different directions ($v_1^i, \ v_2^i, ..., v_{n_i}^i$) but with same $\mathbf{E}[\alpha_{bit}^2]$.

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
