[Supplementary Material]

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

# A    Quantization Details

During the forward pass, each element in a weight or activation tensor $X$ will be quantized as follows:

$$X' = \text{Clamp}(X, q_0, q_{2^k-1}),$$

$$X^I = \lfloor \frac{X' - q_0}{\Delta} \rceil, \text{ where } \Delta = \frac{q_{2^k-1} - q_0}{2^k - 1},$$

$$Q(X) = \Delta X^I + q_0,$$

where $\lfloor \cdot \rceil$ is the round operator, $\Delta$ is the distance between adjacent quantized points, $X^I$ is a set of integer indices, $[q_0, q_{2^k-1}]$ stands for the quantization range of the floating point tensor, and the function Clamp sets all elements smaller than $q_0$ equal to $q_0$, and all elements larger than $q_{2^k-1}$ to $q_{2^k-1}$. It should be noted that $[q_0, q_{2^k-1}]$ can be a subinterval of $[min, max]$, in order to get rid of outliers and better represent the majority of the given tensor. During inference, the expensive floating point arithmetic can be replaced by efficient integer arithmetic for the matrix multiplication with $X^I$, and then followed by a gathered dequantization operation, which will significantly accelerate the computation process. Since we use the quantization-aware fine-tuning scheme, in the backward pass, the Straight-Through Estimator (STE) [4] is used for computing the gradient for $X$.

# B    Search Space

Suppose the number of different mixed-precision settings is $\mathcal{B}$, and the number of different progressive quantization-aware fine-tuning orders is $\mathcal{C}$. The whole search space can be written as $\mathcal{B} \times \mathcal{C}$. We have:

$$\mathcal{B} = m^L.$$

$$\mathcal{C} = \sum_{i=1}^{L} i! \times S(L, i) \to L!.$$

where $m$ is the number of quantization precision options, $L$ is the number of layers in a given model, $S(L, i)$ stands for Stirling numbers of the second kind, which have a growth speed between $O(L!)$ and $O(L^L)$. In the case of layer-wise progressive fine-tuning, where only one layer can be fine-tuned at a time, $\mathcal{C}$ degrades to $L!$.

Given two layers $B_i$ and $B_j$ with average Hessian trace $Tr(B_i)/n_i > Tr(B_j)/n_j$, if we set quantization precision $q_i \geq q_j$, then based on that, we are able to order all $L$ layers in the model according to their average Hessian trace. Considering the situation that $j$ precision options are used out of total number $m$, the mixed-precision problem can be reduced to an integer partition problem, namely, to partition the ordered layers into $j$ different groups, which results in $\binom{j-1}{L-1}$ possible solutions. Since there are $\binom{j}{m}$ different combinations of the $j$ precision options, the total size of search space is $\sum_{j=1}^{m} (\binom{j}{m} \cdot \binom{j-1}{L-1})$.

# C    Extra Results

**Figure 6:** *The top-20 eigenvalues of ResNet50 (3rd/19th layers) and InceptionV3 (4th layer). The dotted lines represent corresponding values of average Hessian trace. As we can see, the trace of Hessian is not dominated by the top-1 eigenvalue, indicating Hessian trace and top-1 eigenvalue to be different metrics.*

**Figure 7:** *Average Hessian trace of different blocks in SqueezeNext and RetinaNet, along with the loss landscape of block 3 and 108 in SqueezeNext, and block 1 and 19 in RetinaNet. It should be noted that block 1 to block 17 in RetinaNet are the ResNet50 backbone, block 18 to block 20 are FPN, and block 21 and block 22 are classification and regression head, respectively. As one can see, the average Hessian trace is significantly different for different blocks. We assign higher bits for blocks with larger average Hessian trace, and fewer bits for blocks with smaller average Hessian trace. For reference, in Figure 2 we showed a similar plot but for InceptionV3 and ResNet50.*

**Figure 8:** *(Left) Average Hessian trace w.r.t. activations in RetinaNet. As we can see, the average Hessian trace varies significantly across activations of different blocks. We use this information to perform mixed-precision activation quantization as discussed in § 2.2. (Right) we show the relationship between the convergence of Hutchinson and the number of data points used for trace estimation on block 5 in RetinaNet. We used 128 data points with 50 Hutchinson steps to plot the left figure.*

**Table 4:** *Comparison between* HAWQ-V2 *and HAQ on ResNet50, InceptionV3 and SqueezeNext. HAQ method can take an order of magnitude more time to find the right bit precision configuration, while our fully automatic approch* HAWQ-V2 *can still achieve significantly higher accuracy with better stability.*

| Model | Method | Top-1 Accuracy | W-Comp | Size(MB) | Search Time(hours) | Speed Up |
|---|---|---|---|---|---|---|
| ResNet50 | Baseline Model | **77.39** | 1.00× | 97.8 | | |
| ResNet50 | HAQ [29] | 75.30 | 10.57× | 9.22 | 10 | |
| ResNet50 | HAWQ-V2 | **75.92** | 12.24× | 7.99 | **0.5** | **> 20×** |
| InceptionV3 | Baseline Model | **77.45** | 1.00× | 91.2 | | |
| InceptionV3 | HAQ [29] | 71.60 | 10.00× | 9.12 | 50 | |
| InceptionV3 | HAWQ-V2 | **75.98** | 12.04× | 7.57 | **0.4** | **> 125×** |
| SqueezeNext | Baseline Model | **69.38** | 1.00× | 10.1 | | |
| SqueezeNext | HAQ [29] | 65.87 | 10.00× | 1.01 | 50 | |
| SqueezeNext | HAWQ-V2 | **68.68** | 9.40× | 1.07 | **0.9** | **> 55×** |

# D  Additional Ablation Study

In Table 5, we show the effectiveness of using average Hessian trace over top eigenvalues. HAWQ uses a heuristic metric $S_i = \lambda_i/n_i$ to select mixed-precision bitwidths, where $\lambda_i$ is the top eigenvalue of the $i^{th}$ layer and $n_i$ is the parameter size. However, as we can see, only using $\lambda_i$ as sensitivity metric in HAWQ (represented as HAWQ-Ablation) can lead to significant accuracy degradation. In contrast, the theoretically derived metric, average Hessian trace, can achieve better results than HAWQ with the same compression ratio. To achieve fair comparison, we constrain HAWQ-V2 to assign the same quantization precision for layers in the same block of InceptionV3, as what HAWQ did, and we make their activation bit settings to be the same (referred as HAWQ-V2-blockwise).

**Table 5:** *The effectiveness of average Hessian trace. The experiments are for InceptionV3 on ImageNet. We abbreviate quantization bits used for weights as "w-bits," quantization bits used for activations as "a-bits," top-1 testing accuracy as "Top-1," and weight compression ratio as "W-Comp." Here "MP" refers to mixed-precision quantization, and we show the lowest bit-precision used in a mixed-precision setting. Blockwise means to assign the same bitwidth for layers within the same block. Compared to* HAWQ *and* HAWQ-*Ablation,* HAWQ-V2 *can achieve higher compression ratio with higher accuracy.*

| Method | Metric | w-bits | a-bits | Top-1 | W-Comp | Size(MB) |
|---|---|---|---|---|---|---|
| Baseline | NA | 32 | 32 | 77.45 | 1.00× | 91.2 |
| HAWQ | $\lambda_i/n_i$ | 2 MP | 4 MP | 75.52 | 12.04× | **7.57** |
| HAWQ-Ablation | $\lambda_i$ | 2 MP | 4 MP | 73.35 | 10.56× | 8.65 |
| HAWQ-V2-blockwise | $\overline{Tr(H_i)}$ | 2 MP | 4 MP | **75.73** | 12.04× | **7.57** |

# E  Pareto Frontier

In order to generate Figure 4 when the size of $\mathcal{B}$ is non-trivial, we break the network into $L/a$ groups, with each group containing $a$ layers. Then we break the x-axis (model size) of the Pareto frontier plot into $b$ intervals. We start with the first $a$ layers of the network. For each interval on the x-axis, we choose top $t$ configurations that have the lowest overall sensitivity based on Equation 10.

Then we consider the next set of $a$ layers. This is similar to the first step, except that now we need to jointly consider the top $t \times b$ configurations selected for the first $a$ layers. Suppose we have $p$ possible bit configurations for the next $a$ layers, we choose new top $t \times b$ configurations out of all $t \times b \times p$ possible bit configurations for the first $2a$ layers.

This process needs to be performed iteratively for all the $L/a$ groups. Our experiments show that the accuracy is not sensitive to the hyperparameters, and we typically set $t$, $b$, $a$ to be 5, 200, 5, respectively. This process can help us to tackle deep networks that have very large size of $\mathcal{B}$. We can even expand $\mathcal{B}$ to be the whole space $m^L$ ($m$ represents the number of quantization precision options), since in that case, the complexity of this algorithm becomes $(L/a) \times t \times b \times m^a$ where $a$ is a constant number irrelevant to $L$. After we obtain Figure 4, we choose the bit configuration with lowest sensitivity given the target model size.