[Reviews · NeurIPS 2020]

Review 1

Summary and Contributions: This paper suggests that Hessian trace can be a good metric to automate the process to decide the number of quantization bits for each layer unlike previous attempts such as using top Hessian eigenvalue. Some mathematical analysis to support that Hessian trace is better than top Hessian eigenvalue is provided while memory footprint and mode accuracy are compared on several models using ImageNet database. This paper also shows that Hessian trace computations can be simplified by following the Hutchinson's algorithm.

Strengths: - Hessian-related metrics have been widely adopted to present different sensitivity of layers. This paper compares a few different Hessian-related approaches and provides some mathematical analysis to claim why Hessian trace can be considered as a good metric to produce some optimal number of quantization bits. - Experiments are performed by large models such as ResNet50 and RetinaNet.

Weaknesses: - The amount of novelty is quite limited because the main contribution of this paper is just to show that Hessian trace can be a better method compared to top Hessian eigenvalue that has been introduced in HAWQ[7]. - One example that showing why Hessian trace is good is only shown in the initial part of Section 2.1 (100x^2 +y^2 vs 100x^2+99y^2). But what is missing to verify the major claim is to show a few major Hessian eigenvalues in some example models. Suppose that top Hessian eigenvalue is highly dominating in general, then Hessian trace would not show distinguished advantages while computational complexity increases. It is necessary to show empirical results to support major claims. - HAQ and a few other previous works do not rely on top Hessian eigenvalue. Hence, this paper is mainly comparing with HAWQ[7] only. It is necessary to prove that Hessian trace is superior to other methods such as machine learning based ideas, e.g., HAQ[22]. - In Table 1 and 2, comparisons are not fair because many methods assume the fixed number of quantization bits while this paper depends on the relaxed assumption that each layer can have different number quantization bits. It would be necessary to compare experimental results on the same principles and hardware requirements. - It is difficult to understand what authors wanted to claim with Figure 4. Since perturbation is defined as a metric that authors have chosen to minimize, it is natural that AMQ results lie on Pareto frontier lines. What is missing is the relationship between total perturbation and model accuracy.

Correctness: Figure 4, Table 1 and Table 2 need to be fixed to improve fairness on comparison.

Clarity: This paper is well written.

Relation to Prior Work: This paper is mainly compared with HAWQ[7]. Other many previous works need to be compared as well (especially HAQ[22] that follow learning-based approach)

Reproducibility: No

Additional Feedback: Unfortunately, this paper shows limited novelty while most claims are made to be compared with HAWQ[7]. Experimental results to compare with previous works need to be made under the same assumptions and hardware resources. ---------------------- [response to authors' rebuttal] Thanks for addressing my concerns in detail. I raise my score to 6. Please include your answers in the final manuscript.


Review 2

Summary and Contributions: This paper considers a mixed-precision quantization scheme. To address three major limitations of existing works, this paper presents AMQ which 1) theoretically proves that the right sensitivity metric is the average Hessian trace, instead of just top Hessian eigenvalue. 2) develops a Pareto frontier based method for automatic bit precision selection of different layers without any manual intervention. 3) develops the first Hessian based analysis for mixed-precision activation quantization, which is very beneficial for object detection. and achieves new state-of-the-art results for a wide range of tasks.

Strengths: Overall, this paper is organized well and easy to follow. Meanwhile, the results are SOTA as shown in the tables.

Weaknesses: 1) In section 2, to theoretically prove that the right sensitivity metric is the average Hessian trace, the paper assumes that the coefficients are identical for all the eigenvectors (i.e., Eq.(2)). Besides, it also assumes that $||\$||Q(W_i)-W_i||$. In fact, the latter has become a very common and natural metric for mix-precision quantization in practice. Therefore, it’s necessary to include these experimental results in the ablation study. Overall, this paper proposes some improvements baseDelta W_i^*||=||\Delta W_j^*||$ for different layers. In fact, these are very strong assumptions. Therefore, the theoretical analysis is not convincing. Meanwhile, there is no experimental result to directly justify the lemma. 2) This paper utilizes Eq.(9) as the metric for bit allocation. To support the average Hessian trace, a strong baseline should be d on HAWQ and shows superior performance compared to existing works. However, on the one hand, the theoretical analysis is not convincing because of the strong assumptions. On the other hand, without sufficient ablation studies, it is hard to verify the correctness and effectiveness of the proposed sensitivity metric. ################## Post-rebuttal I have read the authors' feedback and the comments from other reviewers. The authors are encouraged to include the answers to my concerns in the final version. I change my overall score to "marginally above the threshold". Some minor comments: - It would be better to conduct an ablation study on the effectiveness of mix-precision activations. - Compared with HAWQ, the "automatic design" only marginally improves the accuracy on ImageNet. Therefore, the motivation of "Automatic Mixed-precision Quantization" has weakened somewhat.

Correctness: I personally think the claims and method are ok.

Clarity: The paper is well written.

Relation to Prior Work: Yes, this paper has included discussions on the differences.

Reproducibility: No

Additional Feedback: ############### Post-rebuttal It would be better to release the training codes to help reproduce the results in this paper (I'm not fully convinced that the performance improvement are completely from the "automatic" scheme, leaving out the training tricks).


Review 3

Summary and Contributions: This is one of the Hessian approaches to determine the precision for each layer of the models to minimize search spaces (compared to manual or RL methods). The methods claim better performance than the previous HAWQ method by taking the average Hessian trace instead of the top Hessian eigenvalue. In addition, the method discussed the Hessian for activations, which is normally considered computationally formidable.

Strengths: The authors provide theoretical proof that the Hessian trace is a good indicator of the layer sensitivity to quantization. The authors also demonstrated efficient approximate methods to calcuate the Hessian trace and the Hessian for activations.

Weaknesses: The improvment in performance from HAWQ to AMQ seems moderate to the reviewer, < 0.5% in ResNet and 0.6% in SqueezeNet, leaving the question that whether this is a progressive update or a disruptive one. It would be good to have HAWQ result for RetinaNet as well in the Table 2.

Correctness: The claims and empirical methods are fairly correct. In 3.1, the authors claim that their method is "significantly" faster than HAQ [22] and DNAS[23] could the authors add some data to support that? And How does the speed compare to HAWQ?

Clarity: The paper is well written.

Relation to Prior Work: Yes, this paper comapared AMQ with HAWQ thoroughly.

Reproducibility: Yes

Additional Feedback: The authors' rebuttal addressed my questions fairly. They responded to my first question that whether ~0.5% should be counted as a breakthrough and explained the other advantages that AMQ has over HAWQ. In addition, the authors also agreed to add the comparison between AMQ and HAWQ on RetinaNet, which enhanced the strength of AMQ. Therefore, I will stand by my original positive score. The reason that I decided not to increase my score is because while the authors provided some missing information, the fundamental points have not changed.


Review 4

Summary and Contributions: The paper proposes an automatic approach to select the quantization precision for both weighs and activations for each layer in a mixed-precision neural network. The approach outperforms prior art, HAWQ, both in theory and in practice.

Strengths: Significance: this work was centered around improving the prior art of HAWQ [7]. It certainly accomplishes so with the automatic selection of bit-depths, weight and activation quantization, as well as the low-complexity trace computation. Theoretical grounding: Lemma 1 gives a solid justification for the proposed algorithm and its advantage over HAWQ. The Pareto frontier approach for selecting bit-depth is also first-principled. Empirical evaluation: the work is validated for a mobile architecture (SqueezeNet) where the effect of low-precision is more differentiable, and for object detection where previous quantization work rarely gets involved.

Weaknesses: Significance: the improvement above HAWQ is incremental: the ImageNet results are marginally (<0.5%) better at the same model size;

Correctness: I do not fully understand assumption 1: why does the fine-tuning perturbation have to lie in direction of the sum of H's eigenvectors (as stated in Eq (2))? Regarding Proof of Lemma 1 (line 122): Since the perturbation Delta W does not go to zero (and is probably precision dependent), it may not be safe to ignore high order terms in the Taylor expansion. In this case additional assumptions on higher -order curvatures may be necessary.

Clarity: Yes.

Relation to Prior Work: Yes.

Reproducibility: Yes

Additional Feedback:

[Author Response · NeurIPS 2020]

We thank all reviewers for their insightful feedback, please see answers inlined below.

**R1: Limited novelty compared to HAWQ, paper is just adding Hes-**
**sian trace.** A: Please note that we are proposing a fully automated approach
whereas HAWQ manually chooses bit precision for each layer. Also, de-
spite being fully automated, our accuracy is still higher than HAWQ. Also
we enabled mixed-precision activation which was not covered by HAWQ
and is beneficial for object detection tasks.

**R1: What if top eig. is highly dominating?** A: That is an excellent
question. We show top 20 Hessian eigenvalues in Fig. 1 for ResNet50 and
InceptionV3. As one can see, the top eig. is not an outlier. Theoretically it
is possible that top eig. be very large, but we did not observe it in practice.
We will clarify this in the paper.

**R1, R3: Need to compare with HAQ.** A: We had indeed included com-
parison with HAQ in Tab. 3 in Appendix (p.12 in Supplementary). We
compared performance for ResNet50, InceptionV3, SqueezeNext. We
found it's hard for HAQ agents to search very complicated models, and for
all cases AMQ achieves higher precision with smaller model size with up
to $100\times$ speedup for finding the bit precision setting.

**Figure 1:** *The top-20 eigenvalues of ResNet50 (3rd/19th layers) and InceptionV3 (4th layer).*

**R1: Tab1-2 comparisons are not fair, some methods use fixed bit.** A: Please note that we do compare with HAQ
and HAWQ which are mixed-precision methods. Also please note that prior work with low precision achieve very low
accuracy. Also comparing with fixed bit precision is standard as HAQ also makes similar comparison (for example they
compare with PACT which is fixed-bit quantization in Table 3 of their paper).

**R1: It is difficult to understand Fig. 4.** A: For a given compressed model size there are huge amount of different bit
settings that obey the ordering based on Hessian sensitivity. Pareto Frontier method is a simple and efficient way to find
the bit precision setting that results in the smallest second-order perturbation, jointly for every model size. We include
detailed explanation of how to generate Fig. 4 in Appendix E.

**R1: Missing relationship between total perturbation and accuracy.** A: That is an excellent point. We performed an
ablation study for SqueezeNext between the setting with high total perturbation (which achieves 67.46% accuracy with
model size 1.09MB) and lowest perturbation (which achieves 68.68% accuracy with model size 1.07MB). As expected,
configuration with lower perturbation results in better accuracy. We followed same training for these two runs. We will
add detailed version of this ablation study for different models to the final version.

**R2: Theory assumes strong assumption that** $||\Delta W_i^*|| = ||\Delta W_j^*||$ A: We apologize for the confusion. Please note
that the lemma still holds for cases where $||\Delta W_i^*|| \neq ||\Delta W_j^*||$ (this will introduce a constant coefficient in Eq. 4). We
only used it in the proof for simplicity. Actually because of the $||\Delta W_i^*|| \neq ||\Delta W_j^*||$ cases, the lemma suggests to use
$\overline{Tr}(H_i)||\Delta W_i^*||_2^2$ to determine the sensitivity, which is the exact expression in Eq 9. We will clarify in final version.

**R2: Strong assumption that coefficients are identical for all the eigenvectors.** A: The assumption can be relaxed
to random coefficients as far as those random coefficients have the same 2nd moment, i.e., $\alpha_{bit}$ can be random variable
for different directions but with same $\mathbf{E}[\alpha_{bit}^2]$. We will clarify this in the final version.

**R2: Necessary to include ablation study with** $||Q(W_i) - W_i||$**.** A: That is an excellent point. We conducted this
ablation study for SqueezeNext (which is a hard task since it is already compact) following exactly the same fine-tuning
settings. Using $||Q(W_i) - W_i||$ results in a final accuracy of 67.83% (1.10MB model size), where as AMQ achieves
68.68% (1.07 MB model size). We will add this ablation result for other models to the final version.

**R3, R4: The improvement in performance seems moderate.** A: While $0.5 \sim 0.6\%$ accuracy gain for ImageNet is
actually considerable but we kindly note that the main point of our paper is a fully automated approach as opposed
to the manual approach of HAWQ. The latter only provides relative sensitivity of layers and requires a data scientist
to manually choose the bit precision for each layer. AMQ is fully automated and still achieves better results. It also
up to $100\times$ faster for its end-to-end time to find the bit precision setting of each layer as compared to HAQ (Table 3
Appendix). We think this is quite significant improvement.

**R3: It would be good to have HAWQ result for RetinaNet as well.** A: RetinaNet with HAWQ achieves 33.5 mAP
with model size 17.9MB while AMQ achieves 34.1 mAP with the same model size. We will add this to Tab. 2.

**R4: Why does the fine-tuning perturbation have to lie in direction of the sum of H's eigenvectors?** A: Since
Hessian should be PSD at convergence, its eigenvectors form a complete space for the parameters.

**R4: Additional assumptions on higher-order curvatures may be necessary.** A: That is correct, and we will clarify
this. We should also mention the prohibitive computational cost of evaluating higher order terms.

[Meta-Review · NeurIPS 2020]

This paper on the surface makes a small change of an existing approach to base neural network weight quantization on the Hessian trace instead of max eigenvalue. This is motivated by the observation that the Hessian should be positive definite if the network is optimized to a local optimum. This change, although superficially minor compared with using the largest eigenvalue (as proposed in [7]) leads to the ability to apply Hutchinson's algorithm to compute an efficient sample based approximation to the trace, which leads to large computational speedups and modest performance improvements. The reviewers were unanimous that the paper is at least marginally above the acceptance threshold.